**Data Availability Statement:** All relevant data are within the manuscript and its Supporting Information files.

# Long-term safety and efficacy of ferric citrate in phosphate-lowering and iron-repletion effects among patients with on hemodialysis: A multicenter, open-label, Phase IV trial

Chien-Te Lee[1], Chin-Chan Lee[2], Ming-Ju Wu[3], Yi-Wen Chiu[4], Jyh-Gang Leu[5], Ming-Shiou Wu[6], Yu-Sen Peng[7], Mai-Szu Wu[8,9,10]*, Der-Cherng Tarng[11]*

1 Division of Nephrology, Department of Internal Medicine, Kaohsiung Chang-Gung Memorial Hospital and Chang Gung University College of Medicine, Taoyuan City, Taiwan, 2 Division of Nephrology, Department of Internal Medicine, Keelung Chang-Gung Memorial Hospital, Keelung City, Taiwan, 3 Division of Nephrology, Department of Internal Medicine, Taichung Veterans General Hospital, Taichung City, Taiwan, 4 Division of Nephrology, Department of Internal Medicine, Kaohsiung Medical University Chung-Ho Memorial Hospital, Kaohsiung City, Taiwan, 5 Division of Nephrology, Department of Internal Medicine, Shin Kong Wu Ho-Su Memorial Hospital, Taipei, Taiwan, 6 Division of Nephrology, Department of Internal Medicine, National Taiwan University Hospital, Taipei, Taiwan, 7 Division of Nephrology, Department of Internal Medicine, Far-Eastern Memorial Hospital, New Taipei City, Taiwan, 8 Division of Nephrology, Department of Internal Medicine, Shuang Ho Hospital, Taipei Medical University, New Taipei City, Taiwan, 9 Department of Internal Medicine, School of Medicine, College of Medicine, Taipei Medical University, New Taipei City, Taiwan, 10 TMU Research Center of Urology and Kidney, Taipei Medical University, Taipei, Taiwan, 11 Division of Nephrology, Department of Medicine, Taipei Veterans General Hospital, Taipei, Taiwan

* maiszuwu@gmail.com (MSW); dctarng@vghtpe.gov.tw (DCT)

## Abstract

### Background

We explored the long-term safety and efficacy of ferric citrate in hemodialysis patients in Taiwan, and further evaluated the iron repletion effect and change of iron parameters by different baseline groups.

### Methods

This was a 12-month, Phase IV, multicenter, open-label study. The initial dose of ferric citrate was administered by patients' clinical condition and further adjusted to maintain serum phosphorus at 3.5–5.5 mg/dL. The primary endpoint was to assess the safety profiles of ferric citrate. The secondary endpoints were to evaluate the efficacy by the time-course changes and the number of subjects who achieved the target range of serum phosphorus.

### Results

A total of 202 patients were enrolled. No apparent or unexpected safety concerns were observed. The most common treatment-emergent adverse events were gastrointestinal-related with discolored feces (41.6%). Serum phosphorus was well controlled, with a mean dose of 3.35±1.49 g/day, ranging from 1.5 to 6.0 g/day. Iron parameters were significantly improved. The change from baseline of ferritin and TSAT were 227.17 ng/mL and 7.53%, respectively (p-trend<0.001), and the increase started to slow down after 3–6 months of

**Funding:** This study was conducted by the funding from Panion & BF Biotech Inc. All authors were the investigators in this study and involved in data analysis, decision to publish and manuscript preparation.

**Competing interests:** The authors have declared that no competing interests exist.

treatment. In addition, the increase trend was found only in patients with lower baseline level of ferritin ($\leq$500 ng/mL) and TSAT (<30%).

## Conclusions

Ferric citrate is an effective phosphate binder with favorable safety profile in ESRD patients. The iron-repletion by ferric citrate is effective, and the increase is limited in patients with a higher baseline. In addition to controlling hyperphosphatemia, ferric citrate also shows additional benefits in the treatment of renal anemia.

## Clinical trial registration

ClinicalTrials.gov ID: NCT03256838; 12/04/2017.

## Introduction

Hyperphosphatemia and anemia are the two most frequent complications encountered in end-stage renal disease (ESRD) patients. Hyperphosphatemia is associated with an increased risk of anemia, mineral and bone disorders, and cardiovascular morbidity and mortality [1–3]. Hemodialysis and peritoneal dialysis are standard modalities for ESRD patients. However, dialysis patients still suffer from hyperphosphatemia due to the average intake of phosphorus in the gastrointestinal tract as high as 900 mg/day [4, 5]. Traditional phosphate binders are used to reduce phosphorus absorption in the gastrointestinal tract, which may lead to adverse results such as hypercalcemia, systemic toxicity, gastroduodenal lesions, or lower adherence rates due to heavy pill burden and inconvenient formulations [6]. As kidney function declines in chronic kidney disease (CKD) progression, the erythropoietin production decreases, and it exacerbates anemic condition due to lowering of iron level. With its additional benefits of iron supplementing for renal anemia, iron-based phosphate binder has become a popular choice for CKD patients [7, 8].

Ferric citrate is safe and efficacious in the management of hyperphosphatemia and anemia in both non-dialysis [9–15] and dialysis-dependent [16–18] CKD patients. All studies showed consistent serum phosphorus control with good tolerability. The most common adverse events (AEs) were feces discoloration and diarrhea. Although ferric iron was traditionally deemed as a non-absorbable form of iron, the long-term studies of ferric citrate showed the ferric iron from the citric chelated structure could still be utilized to improve iron panel and hemoglobin level, and further reduced the need for erythropoiesis-stimulating agents (ESA) and intravenous iron in hemodialysis patients [19–23]. The available long-term data can only be found in the US and Japanese populations which represented two extreme ferritin groups, 887 ng/ml and 73 ng/ml respectively [24, 25]. No study has ever evaluated the long-term effect for other ESRD patient groups with a medium level of ferritin (300~500 ng/mL) such as Taiwanese population. This Phase IV trial designed as observational and non-interventional study evaluated the safety, tolerability, and effectiveness in real-world settings in which concomitant treatments (intravenous iron and ESAs) were not limited. By further analyzing the change of ferritin and transferrin saturation (TSAT) with different baseline levels, this study allows understanding of ferric citrate's iron absorption/utilization and its effects on anemia improvement in ESRD patients with medium level of ferritin, its impacts on existing anemia treatments and long-term safety profile in the clinical practice.

## Materials and methods

### Study design

This study is a multicenter, open-label, Phase IV study to evaluate the long-term safety and efficacy of ferric citrate in subjects with ESRD on hemodialysis. The study was conducted in accordance with the Declaration of Helsinki, and all participants provided written informed consent. The study protocol was approved by the Institutional Review Board the participating centers (Shin Kong Wu Ho-Su Memorial Hospital, #20170101C; Taichung Veterans General Hospital, #SC16169B; Kaohsiung Chang Gung Memorial Hospital, #201700047A3; Keelung Chang Gung Memorial Hospital, #201700047A3; Far Eastern Memorial Hospital, #105086-I; Taipei Medical University Hospital, #N201612068; Kaohsiung Medical University Chung-Ho Memorial Hospital, #KMUHIRB-F(I)-20170004; National Taiwan University Hospital, #201701034MSA; and Taipei Veterans General Hospital, #2017-02-003AU). Written informed consent was obtained from each participant. All procedures performed in the present study were in accordance with Declaration of Helsinki. The study was registered on ClinicalTrials. gov (NCT03256838; 12/04/2017), and the clinical study protocol was shown in S1 Appendix. All participants received the standard care for hemodialysis according to routine hospital practice, except for pharmacological interventions for hyperphosphatasemia. There was no wash-out period between previous medications/therapies and this study medication. Ferric citrate (500 mg/capsule containing 105 mg of ferric iron) was taken with meals for 12 months. For subjects whose previous phosphate binder doses were equivalent to less than and greater than 4.5 g/day of calcium-based phosphate binder, the initial doses of ferric citrate were 3 g/day and 4.5 g/day, respectively. During the study, the dose of ferric citrate was further titrated by the investigators to achieve the goal of maintaining serum phosphorus levels between 3.5 and 5.5 mg/dL, with a maximum dose of 12 g/day.

The study period included an enrollment visit (baseline, M0), routine monthly visits during the 12-month treatment period (M1, M2, M3, etc.), and a follow-up visit one month after the end of treatment (EOT). EOT was defined as the final treatment visit or early termination of treatment. In addition to physical examination, vital signs, essential hematology, and biochemistry tests were performed at each visit. The use of concomitant medications such as vitamin D, intravenous iron preparations, or ESA was not limited during the study.

### Subject population

From April 2017 to September 2019, a total of 224 subjects were screened for eligibility. The inclusion criteria included (1) age > 18 years and provision of written informed consent; (2) ESRD patients undergoing hemodialysis 3 times/week, and necessary to receive medication for hyperphosphatemia; (3) serum ferritin < 1,000 ng/mL and TSAT < 50% at the time of enrollment; and (4) women of child-bearing potential were willing to use contraception during the study period. Patients who met any of the following criteria were excluded from this study. (1) Patients had any known contraindication to ferric citrate, including but not limited to allergy to ferric citrate; hypophosphatemia; hemochromatosis or iron overload syndrome; and active severe gastrointestinal disease. (2) Patients underwent parathyroidectomy or percutaneous ethanol injection therapy within 3 months prior to enrollment visit or patients had serum calcium < 7 mg/dL at the enrollment visit. (3) Patients participated in another interventional study within 30 days prior to the enrollment. (4) The female patients were currently pregnant or breastfeeding. (5) Patients with unstable medical conditions or psychiatric conditions and were not suitable for this study based on the investigator's judgment.

## Safety and efficacy assessments

The safety endpoint was to evaluate the safety profile, including treatment-emergent adverse events (TEAEs), clinical laboratory evaluations, vital sign measurements, physical examination, and 12 lead electrocardiograms. The efficacy was evaluated by the time-dependent change in serum phosphorus levels and the proportion of achieving the target range of serum phosphorus (3.5–5.5 mg/dL) throughout the treatment. Additional exploratory study endpoints included serum calcium, iron, ferritin, TSAT, total iron binding capacity (TIBC), hemoglobin, intact parathyroid hormone (iPTH), and the dose change of intravenous iron and ESA.

All subjects who received at least one dose of study medication were evaluated as the safety population. The full analysis set (FAS) was defined as the participants who satisfied all eligibility criteria, took at least one dose of study medication, and had at least one post-treatment evaluation for efficacy.

## Statistical analyses

Continuous data were expressed as mean and standard deviation (SD), and categorical data were expressed in numbers and percentages in the analysis of baseline characteristics and TEAEs. The values of serum phosphorus, iron-, anemia-, and bone-mineral-related parameters were expressed as mean and standard error (SE). P-trend was calculated by applying a general linear model for all continuous variables. P-value was used to compare the M0 and EOT by paired t-test. Statistical analyses were performed by SPSS software version 22.0 (IBM, Armonk, NY, USA), and a two-sided $p < 0.05$ was considered statistically significant.

## Results

### Participant characteristics

A total of 224 subjects were screened for eligibility, of which 202 subjects were enrolled and 117 subjects (57.9%) completed all the scheduled ferric citrate treatment for 12 months (Fig 1). All 202 subjects, received at least one dose of study medication, were included in the safety population, and 197 patients were evaluated as FAS. The baseline demographic characteristics of the safety population are shown in Table 1. The study group comprised 53.5% male, and the mean age was 60.9 ± 9.8 years, with a mean bodyweight of 64.9 ± 14.1 kg. The mean duration of hemodialysis was 8.2 ± 6.6 years. Patients undergoing hemodialysis for > 5 years were 59.9%. The main causes of renal disease were diabetic nephropathy (33.7%) and hypertensive nephrosclerosis (25.7%). The most common comorbidities were hypertension (69.8%), diabetes mellitus (40.6%), and hyperlipidemia (31.2%). There were 94.1% subjects who received ESA, 56.9% who received intravenous iron, and 52.5% who received vitamin D before the study. Calcium-based phosphate binder was the most common prior to the study medication (82.7%), and other prior use of phosphate binders included aluminum-based (19.3%), sevelamer-based (7.4%), lanthanum-based (7.9%), and ferric citrate-based binders (5.9%).

### Safety

The safety was evaluated based on the physical examinations, changes in biochemical laboratory parameters, and profile of TEAEs. No clinically significant fluctuations in physical examinations and laboratory parameters including aluminum, sodium, potassium, albumin, alanine aminotransferase (ALT), and aspartate aminotransferase (AST) were observed during the study (S1 Table).

There were no drug-related deaths during the study period. The most common TEAEs with an incidence rate ≥ 20% reported by System Organ Class (SOC) were gastrointestinal

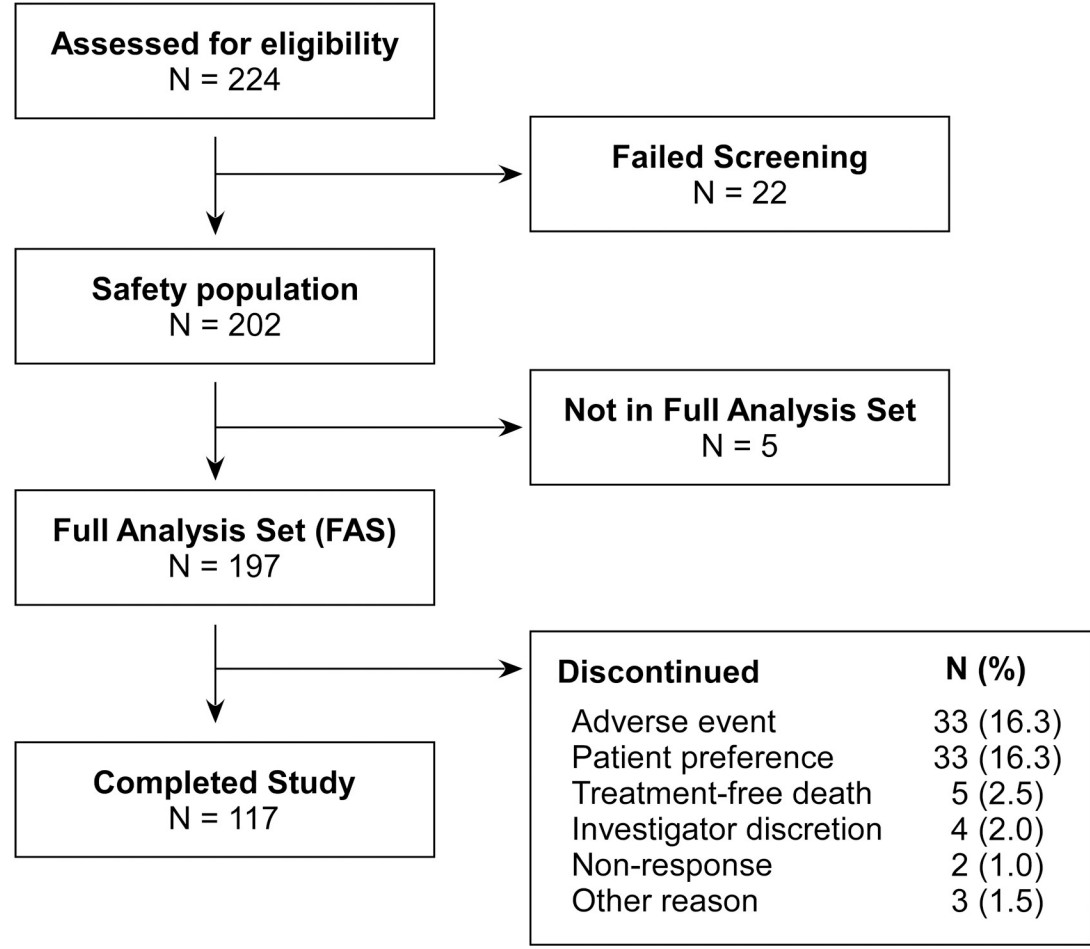

**Fig 1. Flow diagram of subjects by completion status.**

disorders, infections and infestations, injury, poisoning, and procedural complications, respiratory, thoracic and mediastinal disorders, musculoskeletal and connective tissue disorders, and skin and subcutaneous tissue disorders (Table 2). Discolored feces (41.6%) was the most common TEAE. Other common TEAEs with an incidence rate ≥ 10% reported by preferred term were cough, diarrhea, and constipation. Most of these TEAEs were mild in severity. The most frequently reported serious TEAEs, including pneumonia (5.9%), acute myocardial infarction (3.0%), cardiac failure congestive, and coronary artery disease (2.0% each, respectively) (Table 3). None of the serious TEAEs were suspected to be related to ferric citrate treatment. S2 Table showed the most common drug-related TEAEs with discolored feces (41.1%) in the majority. Most TEAEs that led to drug discontinuation were mild to moderate, mainly related to gastrointestinal disorders (S3 Table). Overall, none of the serious TEAEs led to drug discontinuation, suggesting that ferric citrate is safe for ESRD patients.

## Efficacy

Serum phosphorus level was well controlled and maintained throughout the study. The mean serum phosphorus was 5.38 mg/dL at M0 and 5.19 mg/dL at EOT with a significant overall decreasing trend from M0 to EOT (P-trend = 0.003) (Table 4). There was a slight increase in the serum phosphorus level in the first month of study due to the monthly titration design

**Table 1. Demographic and baseline characteristics of study population (N = 202).**

| Baseline characteristics | | | |
|---|---|---|---|
| Age (year) | | 60.9 ± 9.8 | |
| Gender | | | |
| | Male | 108 | (53.5%) |
| | Female | 94 | (46.5%) |
| Body weight (kg) | | 64.9 ± 14.1 | |
| Age at diagnosis of ESRD (year) | | 49.7 ± 12.0 | |
| History of hemodialysis (year) | | 8.2 ± 6.6 | |
| History of hemodialysis | | | |
| | ≤ 1 year | 10 | (5.0%) |
| | 1–5 years | 71 | (35.1%) |
| | 5–10 years | 52 | (25.7%) |
| | >10 years | 69 | (34.2%) |
| Primary etiology | | | |
| | Diabetes nephropathy | 68 | (33.7%) |
| | Hypertensive nephrosclerosis | 52 | (25.7%) |
| Comorbidity [a] | | | |
| | Hypertension | 141 | (69.8%) |
| | Diabetes mellitus | 82 | (40.6%) |
| | Hyperlipidemia | 63 | (31.2%) |
| Prior use of medication [a],* | | | |
| | ESA | 190 | (94.1%) |
| | Intravenous iron | 115 | (56.9%) |
| | Vitamin D | 106 | (52.5%) |
| Prior use of phosphate binder [a],* | | | |
| | Calcium-based binders | 167 | (82.7%) |
| | Aluminum-based binders | 39 | (19.3%) |
| | Sevelamer-based binders | 15 | (7.4%) |
| | Lanthanum-based binders | 16 | (7.9%) |
| | Ferric citrate-based binders | 12 | (5.9%) |

Continuous variables were presented as mean ± standard deviation, and categorical variables were presented as number and percentage.

[a] Multiple entries allowed.

* Within 3 months before to the study.

Abbreviations: ESRD, end-stage renal disease; ESA, erythropoiesis stimulating agents.

from a low starting dose without washout (Fig 2A). However, the subsequent serum phosphorus levels were stably maintained within the target range of 3.5–5.5 mg/dL with an achievement rate of 64.4% at M6 and 53.6% at EOT.

With overall treatment adherence of 83.02%, patients achieved the efficacy target with a mean daily dose of 3.35 ± 1.49 g/day. A majority (> 90%) of the patients received a dose of under 6 g/day and > 80% of patients received a dose ranging between 1.5~6 g/day.

## Changes of hematological and biochemical laboratory parameters

Hemoglobin and serum iron levels increased significantly and reached the highest levels after 3 and 6 months of ferric citrate treatment, respectively (Fig 2B, P-trend < 0.001). Serum ferritin and TSAT level increased gradually while TIBC showed a decreased trend, and the changes

**Table 2. TEAEs with incidence rate ≥ 20% by SOC (N = 202).**

| TEAEs by SOC and PT | N | (%) |
|---|---|---|
| Gastrointestinal disorders | | |
| Discolored feces | 84 | (41.6%) |
| Diarrhea | 28 | (13.9%) |
| Constipation | 26 | (12.9%) |
| Abdominal pain | 18 | (8.9%) |
| Abdominal distension | 13 | (6.4%) |
| Gastroesophageal reflux disease | 10 | (5.0%) |
| Infections and infestations | | |
| Upper respiratory tract infection | 18 | (8.9%) |
| Pneumonia | 14 | (6.9%) |
| Viral upper respiratory tract infection | 14 | (6.9%) |
| Injury, poisoning and procedural complications | | |
| Arteriovenous fistula site complication | 20 | (9.9%) |
| Shunt malfunction | 13 | (6.4%) |
| Respiratory, thoracic and mediastinal disorders | | |
| Cough | 33 | (16.3%) |
| Musculoskeletal and connective tissue disorders | | |
| Pain in extremity | 14 | (6.9%) |
| Back pain | 11 | (5.4%) |
| Musculoskeletal pain | 10 | (5.0%) |
| Skin and subcutaneous tissue disorders | | |
| Pruritus | 19 | (9.4%) |

Data was presented as number and percentage, and the N was based on the number of patients experiencing ≥ 1 TEAE, not the number of events.

Abbreviations: TEAEs, treatment-emergent adverse events; SOC, system organ class; PT, preferred term.

reached a plateau at M3 and maintained till the EOT (Fig 2C, P-trend < 0.001). The calcium level gradually decreases (P-trend = 0.02) and the iPTH levels gradually increased (P-trend = 0.03), as shown in Fig 2D.

**Table 3. Most common serious TEAEs with incidence rate >1% (N = 202).**

| Serious TEAEs by PT | N | (%) |
|---|---|---|
| Pneumonia | 12 | (5.9%) |
| Acute myocardial infarction | 6 | (3.0%) |
| Cardiac failure congestive | 4 | (2.0%) |
| Coronary artery disease | 4 | (2.0%) |
| Acute pulmonary edema | 3 | (1.5%) |
| Angina unstable | 3 | (1.5%) |
| Cellulitis | 3 | (1.5%) |
| Hyperparathyroidism | 3 | (1.5%) |
| Pulmonary oedema | 3 | (1.5%) |

Data was presented as number and percentage, and the N was based on the number of patients experiencing ≥1 TEAE, not the number of events. Abbreviations: TEAEs, treatment-emergent adverse events; PT, preferred term.

**Table 4. Changes in serum phosphorus, hemoglobin, iron-related parameters, calcium and iPTH (N = 197).**

| | M0 | M3 | M6 | M9 | EOT | P-trend |
|---|---|---|---|---|---|---|
| Serum phosphorus (mg/dL) | 5.38 ± 0.09 | 5.16 ± 0.10 | 5.16 ± 0.11 | 5.12 ± 0.11 | 5.19 ± 0.11 | 0.003 |
| Hemoglobin (g/dL) | 10.65 ± 0.09 | 11.15 ± 0.12 | 11.11 ± 0.12 | 11.06 ± 0.12 | 11.03 ± 0.12 | <0.001 |
| Serum iron (μg/dL) | 64.48 ± 1.57 | 80.36 ± 2.46 | 86.02 ± 3.09 | 81.93 ± 3.34 | 75.64 ± 2.03 | <0.001 |
| Ferritin (ng/mL) | 353.77 ± 16.00 | 481.49 ± 28.47 | 588.41 ± 26.70 | 619.98 ± 26.70 | 581.37 ± 27.91 | <0.001 |
| TSAT (%) | 24.11 ± 0.87 | 33.07 ± 1.36 | 34.17 ± 1.75 | 33.23 ± 1.85 | 32.36 ± 1.24 | <0.001 |
| TIBC (μg/dL) | 238.59 ± 3.42 | 216.96 ± 3.23 | 219.65 ± 4.03 | 212.57 ± 3.26 | 212.07 ± 2.93 | <0.001 |
| Serum calcium (mg/dL) | 9.57 ± 0.06 | 9.51 ± 0.07 | 9.46 ± 0.07 | 9.35 ± 0.08 | 9.32 ± 0.06 | 0.02 |
| iPTH (ng/L) | 453.25 ± 32.36 | 486.75 ± 36.05 | 575.77 ± 39.26 | 613.78 ± 48.76 | 555.37 ± 35.41 | 0.03 |

Data was presented as mean ± standard error. P-trend values were based on variable containing mean value.

Abbreviations: M0, baseline; M3, 3 months; M6, 6 months; M9, 9 months; EOT, end of treatment; TSAT, transferrin saturation; TIBC, total iron-binding capacity; iPTH, intact parathyroid hormone.

## Changes in demand for intravenous iron and ESA administration

The required dose of intravenous iron was gradually decreased from 0.63 mg/day at the first quarter (1–3 months) to 0.16 mg/day in the fourth quarter (9–12 months), showing a 74.6%

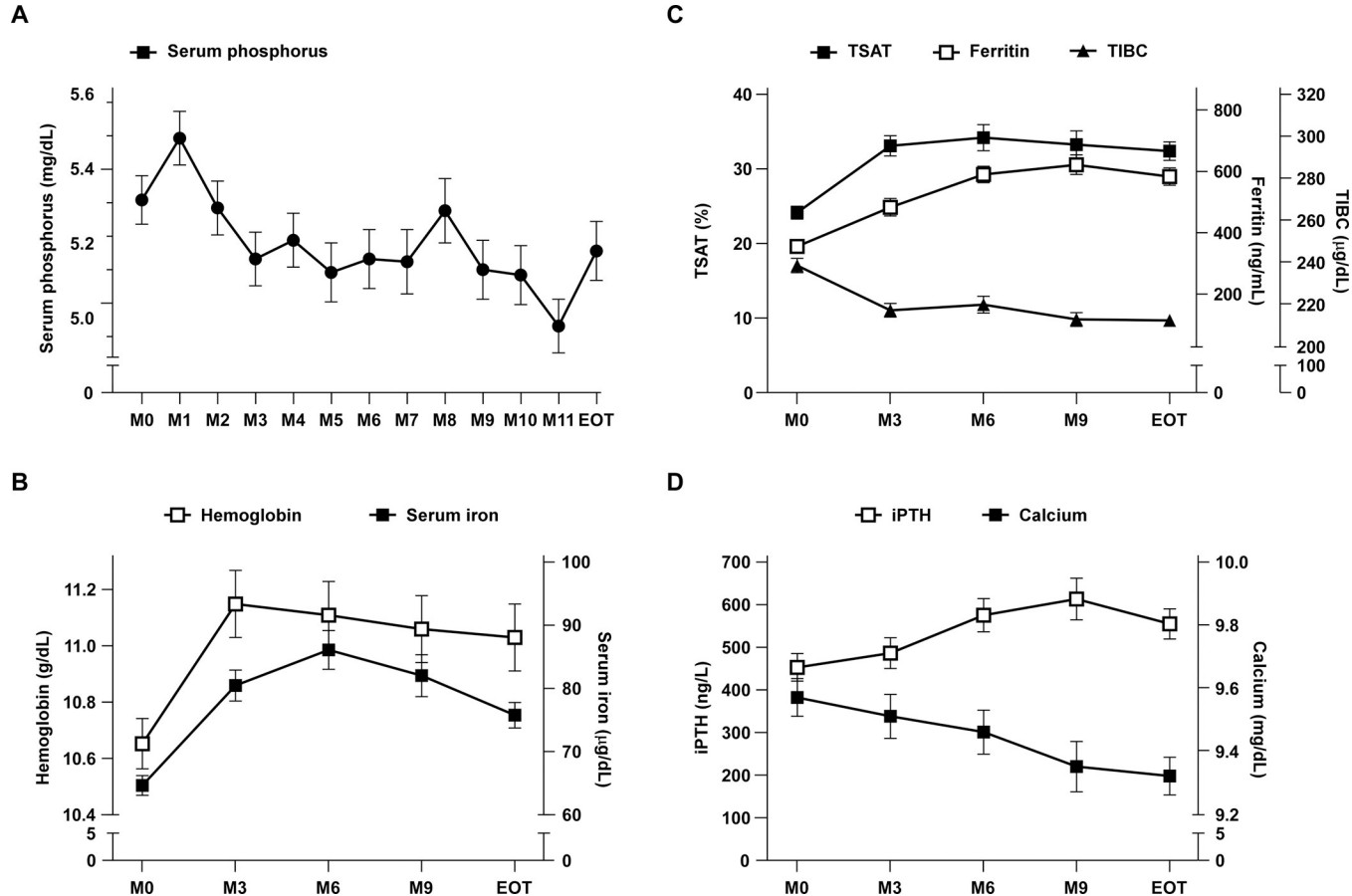

**Fig 2.** The impact of ferric citrate treatment on the changes of (A) serum phosphorus, (B) hemoglobin and serum iron, (C) TSAT, ferritin, and TIBC and (D) iPTH and calcium from M0 to EOT. Abbreviations: TSAT, transferrin saturation; TIBC, total iron binding capacity; iPTH, intact parathyroid hormone; M0, baseline; EOT, end-of-treatment. The values were expressed as mean and standard error (SE).

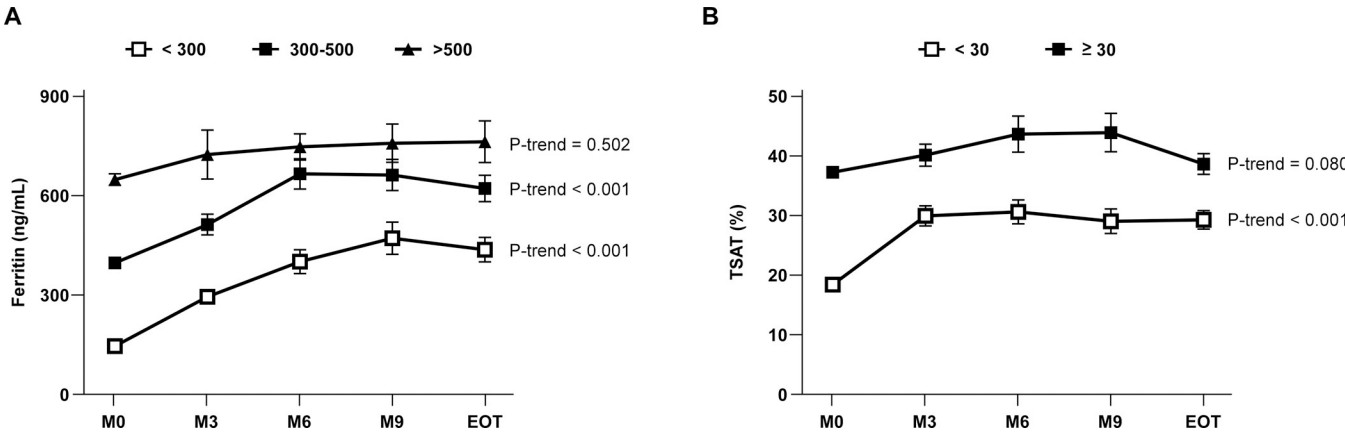

**Fig 3.** Changes in (A) ferritin and (B) TSAT from M0 to EOT according to different baselines of ferritin (<300 ng/mL, 300–500 ng/mL, >500 ng/mL) and TSAT (<30%, ≥30%). The trend of continuous data was calculated by a linear regression. Abbreviations: TSAT, transferrin saturation; M0, baseline; EOT, end-of-treatment. The values were expressed as mean and standard error (SE).

reduction over the 12-month observation (S1 Fig). In addition, the proportion of ESRD patients requiring intravenous iron administration was gradually decreased from 23.3% to 6.5%. Although the proportion of ESRD patients requiring ESA treatment has not changed obviously, the required ESA dose was decreased from 597.2 IU/day at the first quarter to 465/1 IU/day in the fourth quarter, showing a 22.1% reduction over the 12 months of ferric citrate treatment.

## Changes of ferritin and TSAT by baseline level

A *post hoc* subset analysis was performed to stratify ferritin (<300 ng/mL, 300–500 ng/mL, >500 ng/mL) and TSAT (<30%, ≥30%) by baseline levels. The results shown in Fig 3 indicated that the increase was greater in subjects with lower baseline level for ferritin <300 ng/mL and 300–500 ng/mL, which changed from 144.53 ± 8.91 ng/mL to 437.42 ± 37.17 ng/mL and 397.55 ± 7.92 ng/mL to 623.22 ± 39.98 ng/mL respectively (P-trend < 0.001) and TSAT <30%, which changed from 18.45 ± 0.83% to 29.35 ± 1.56% (P-trend < 0.001), however did not alter significantly in higher baseline subsets for ferritin >500 ng/mL and TSAT ≥30% which changed from 650.14 ± 17.42 ng/mL to 765.17 ± 63.31 ng/mL (P-trend = 0.502) and 37.36 ± 0.64% to 38.74 ± 1.75% (P-trend = 0.08) respectively.

## Effect of vitamin D concomitant treatment on iPTH level

Despite serum phosphorus levels was controlled during the study period, iPTH levels gradually increased. The use of the concomitant vitamin D was further analyzed (S4 Table) and it was found no significant change in iPTH level for subjects prescribed with vitamin D treatment (P = 0.189). For subjects without vitamin D treatment, the iPTH level was significantly increased from 323.82 ± 40.98 to 472.94 ± 47.74 ng/L (P<0.001).

## Discussion

This is a multicenter, open-label, Phase IV study to evaluate the long-term safety and efficacy of ferric citrate for phosphate-lowering effect in ESRD patients on maintenance hemodialysis. Predictably, ferric citrate, at a mean daily dose of 3.35 ± 1.49 g/day, was well-tolerated during the 12-month treatment and efficaciously controlled serum phosphorus levels (3.5–5.5 mg/dL). Furthermore, the levels of serum iron, ferritin, and TSAT, as well as hemoglobin

significantly increased during the treatment, while the level of TIBC decreased. Although the intravenous iron and ESAs were not limited in this study, the demands for these concomitant medications were reduced. No drug-related death or serious TEAEs were observed during 12-month treatment. No new or unexpected safety concerns were observed. The common TEAEs occurred in the gastrointestinal tract. Discolored feces, diarrhea, and constipation were the most common gastrointestinal disorders. The results of this study indicated that ferric citrate is a safe and effective phosphate binder to control hyperphosphatemia with good tolerability, and effectively to improve iron utility and stores in real-world settings.

This study showed consistent efficacy and safety results with previous studies including a placebo-controlled study in Taiwanese [26], long-term studies in Japanese [18, 27], and a 52-week active-controlled followed by a 4-week placebo-controlled study in the United States and Israel [16]. In a previous 8-week study of hemodialysis patients in Taiwan [26], daily dose of 4–6 g was well-tolerated. After 8 weeks of treatment, serum phosphorus declined significantly, while ferritin levels increased significantly. In addition to confirming long-term safety of efficacy, this present study further revealed the different iron repletion effects of ferric citrate in ESRD patients with high, medium and low baseline ferritin levels. As expected, that the bodyweight, size, and diet habits were similar in Asia, the average dose used to maintain serum phosphorus target was similar at around 3 g/day in Taiwanese and Japanese patients. However, the baseline ferritin and TSAT levels in the Japanese ESRD population were much lower [25], and that of Taiwanese ESRD patients are generally maintained at ferritin level of 300−500 ng/mL and TSAT of 30%−50% which are similar to the anemia management suggested in other Asian countries and Europe [28]. This study provided a practical direction to physicians to use ferric citrate in patients with a medium level of ferritin and TSAT.

The treatment of ferric citrate improved iron utility and stores. Pergola et al. [15] and Komatsu et al. [14] recently reported that ferric citrate is also safe and effective to treat iron deficiency anemia with and without nondialysis-dependent CKD, respectively. Long-term use of ferric citrate in nondialysis-dependent CKD has no apparent detrimental effect on kidney functions. In this present study, the ingested iron is absorbed for hematopoiesis as reflected from an average increase in hemoglobin level around 0.5 g/dL in ESRD patients, and the hemoglobin level was maintained until the end of the study, despite the reduced uses of intravenous iron and ESA. The increase of iron storage was saturated when TSAT reached a plateau level and the rate of increase in ferritin slowed down after 6 months of treatment. Subset analysis of ferritin and TSAT showed that the increase was greater in patients with lower baseline levels but did not alter significantly in higher baseline subsets (ferritin >500ng/mL, TSAT $\geq$ 30%). Unlike traditional intravenous iron injections, oral ferric citrate is under endogenous regulatory control in the gastrointestinal tract. The regulation from hepcidin and membrane proteins on the small intestine mucosal cell such as DcytB (Ferrireductase), DMT1 (divalent metal transporter1), and Fpn1 (Ferroportin1) tightly control iron's uptake and mobilization into the blood stream [29]. This mechanism allows iron absorption as needed, and lowers the chance of excessive absorption when the iron stores in the body are sufficient.

As ferric citrate is simultaneously improving the iron stores and hemoglobin level, the dose and demand for intravenous iron and ESA in ESRD patients are reduced accordingly. Anemia is frequently encountered in CKD patients and prevalence rate increases along with CKD progression. As kidney function declines, it negatively affects erythropoietin production and iron absorption, therefore leading to chronic renal anemia. Kidney Disease: Improving Global Outcomes (KDIGO; https://kdigo.org/guidelines/) proposed the clinical guidelines for the management of anemia and iron supplementation in CKD patients [22]. However, the use of high doses of ESA is known to be related with an increased risk of stroke [30, 31], hypertension [32], vascular access thrombosis [33], and cardiovascular disease [20, 34–37]. High doses of

ESA may also lead to functional iron deficiency in CKD and ESRD patients. Similarly, intravenous iron administration is also associated with risk of thrombosis, infection, allergic and non-allergic reactions, and cardiovascular death [19, 21, 23]. Intravenous iron aggravated oxidative stress, increased atherogenesis and cardiovascular toxicity, and increased the tendency for infections [38, 39]. It is worth noting that in this study, ferric citrate gradually reduced the demand (72.1% reduction) and the dose (74.6% reduction) of intravenous iron in ESRD patients. The ESA dose was also reduced by 22.1% after 12-month of ferric citrate treatment, while the demand for ESA was not decreased dramatically. This dose reduction effect was also reported by Umanath et al. while hemoglobin was stably maintained [17]. In the pharmacoeconomic aspect, fewer intravenous injections may reduce the shortage of nurses, overall medical expenses, and the risks of side effects such as infections and allergies. However, the benefits in iron repletion and reduction of concurrent anemia treatments were only observed in ferric citrate but not in other iron-based binders. In a 52-week, Phase III study by Covic et al. [40], both sevelamer carbonate and sucroferric oxyhydroxide did not show an increase in iron stores and a reduction in the dose or need for ESA and intravenous iron [11, 17].

In this study, iPTH levels increased despite a significant reduction in serum phosphorus and calcium levels. As secondary hyperparathyroidism is also a common complication in CKD patients, reducing dietary phosphorus uptake along with the use of vitamin D analogs and calcimimetics is usually recommended by KDIGO and other guidelines. In a retrospective study by Yoshida et al. [18], more than 93% of patients concomitantly prescribed vitamin D during the 36-month observation period and no significant change in iPTH was found with ferric citrate treatment. In our present study, only 50.3% of patients in this study concomitantly prescribed vitamin D (S4 Table), suggesting that vitamin D may play a role in controlling the iPTH levels in ESRD patients. Further prospective study is required to clarify this issue.

This study presented long-term clinical experience for up to one year in real-world setting, and some limitations were acknowledged. First, there was a lack of intervention, and it was limited with bias by the missing values resulting from patients who did not attend all scheduled visits. Second, 5.9% of patients had taken ferric citrate prior to the study. These patients may have a lower risk of adverse events during the study period, which may increase bias. Third, the safety and efficacy analyses are limited to 12 months of treatment; therefore, the conclusions of this study can only be limited within this time frame. Nonetheless, this Phase IV study proves that 12-month ferric citrate treatment is safe and effective in the control of serum phosphorus in hemodialysis patients in Taiwan with the additional benefit of iron-repletion.

## Conclusion

Long-term ferric citrate treatment is safe and well-tolerated for ESRD patients with hyperphosphatemia, with a dose range of 1.5–6.0 g/day in Taiwanese population. Ferric citrate can simultaneously replete iron, increase hemoglobin levels, and reduce the dose and/or demand of ESA and intravenous iron. There is no obvious safety concern during 12-month ferric citrate treatment. In routine clinical practice, the dosage of ferric citrate can be adjusted based on serum phosphorous levels and iron-related parameters. Overall, ferric citrate as a phosphate binder has shown promising iron-replenishing benefit for the long-term treatment of hyperphosphatemia in ESRD patients and well-tolerated.

## Supporting information

**S1 Checklist. CONSORT 2010 checklist of information to include when reporting a randomised trial***.
(DOC)

**S2 Checklist. TREND statement checklist.**
(PDF)

**S1 Table. Changes of clinical laboratory parameters during the treatment period.**
(DOCX)

**S2 Table. Most common drug-related TEAEs with incidence rate > 1%.**
(DOCX)

**S3 Table. Most common TEAEs leading to drug discontinuation with incidence rate >1%.**
(DOCX)

**S4 Table. Changes of iPTH levels after stratified by concomitant vitamin D treatment.**
(DOCX)

**S1 Fig. Ferric citrate reduces the need for intravenous iron and ESA.** (A) Intravenous iron dosage and administration. (B) ESA dosage and use.
(JPG)

**S1 Appendix. Clinical study protocol.**
(PDF)

## Author Contributions

**Conceptualization:** Chien-Te Lee, Mai-Szu Wu, Der-Cherng Tarng.

**Formal analysis:** Chien-Te Lee, Mai-Szu Wu, Der-Cherng Tarng.

**Investigation:** Chien-Te Lee, Chin-Chan Lee, Ming-Ju Wu, Yi-Wen Chiu, Jyh-Gang Leu, Ming-Shiou Wu, Yu-Sen Peng, Mai-Szu Wu, Der-Cherng Tarng.

**Writing – original draft:** Chien-Te Lee, Mai-Szu Wu, Der-Cherng Tarng.

**Writing – review & editing:** Chien-Te Lee, Chin-Chan Lee, Ming-Ju Wu, Yi-Wen Chiu, Jyh-Gang Leu, Ming-Shiou Wu, Yu-Sen Peng, Mai-Szu Wu, Der-Cherng Tarng.

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
