## [Decision Letter · Decision Letter 0]

26 Oct 2021

PONE-D-21-25691Long-term safety and efficacy of ferric citrate in phosphate-lowering and iron-repletion effects among patients with on hemodialysis: a multicenter, open-label, Phase IV trialPLOS ONE

Dear Dr. TARNG

Thank you for submitting your manuscript to PLOS ONE. After careful consideration, we feel that it has merit but does not fully meet PLOS ONE’s publication criteria as it currently stands. Therefore, we invite you to submit a revised version of the manuscript that addresses the points raised during the review process.

We look forward to receiving your revised manuscript.

Kind regards,

Pasqual Barretti, Ph.D., MD

Academic Editor

PLOS ONE

Journal Requirements:

"This study was sponsored by Panion & BF Biotech Inc. All authors were the investigators in this study and involved in the manuscript preparation in collaboration with Panion & BF Biotech Inc."

"The funders had no role in study design, data collection and analysis, decision to publish, or preparation of the manuscript"

Additional Editor Comments (if provided):

This study needs a deep revision from the authors to be considered fro publication. The reviewers have distinct approaches and analysis, all of the very relevant. The reviewer 1 has raised concerns about the relevance and the design, while the reviewer has made very important questions about the statistical analysis. The reviewer 3 has focused in the improvement of the discussion on the explanations for your results.So, may decision is "major revision "

Reviewers' comments:

Reviewer's Responses to Questions

**Comments to the Author**

1. Is the manuscript technically sound, and do the data support the conclusions?

Reviewer #1: Yes

Reviewer #2: Yes

Reviewer #3: Partly

2. Has the statistical analysis been performed appropriately and rigorously? 

Reviewer #1: Yes

Reviewer #2: Yes

Reviewer #3: No

3. Have the authors made all data underlying the findings in their manuscript fully available?

Reviewer #1: Yes

Reviewer #2: Yes

Reviewer #3: Yes

4. Is the manuscript presented in an intelligible fashion and written in standard English?

Reviewer #1: Yes

Reviewer #2: Yes

Reviewer #3: Yes

5. Review Comments to the Author

Reviewer #1: Lee et al. examined the long-term safety and efficacy of ferric citrate in patients undergoing hemodialysis. They showed that ferric citrate efficaciously and safely decreased serum phosphate levels in a prospective, multi-center, open-label, one-year study.

There are a lot of previous studies showing that ferric citrate efficaciously and safely decreased serum phosphate levels. However, many of them have not been cited in this manuscript. The novelty of the present study is questionable.

As described in the Methods section, 5.9% of patients had taken ferric citrate prior to the study. These patients were very low risk of adverse events in the study period. The study has been biased on this point.

In the Discussion section, the authors described that “the safety and efficacy analyses are limited to 12 months of treatment; therefore, the conclusions of this study can only be limited within this time frame” as a limitation of this study. Regarding this point, a 3-year retrospective study (Yoshida et al., Int Urol Nephrol, 2021) has been reported recently.

Please correct the style of references.

Multiple names of adverse events have been written using the capital letters. What do they mean?

Reviewer #2: Phosphate binders and iron supplementation are a part of the basic treatment of HD patients. Phosphate binders bind phosphorus in the gut and prevent their absorption.

Iron based phosphate binders are a new line of treatment. In HD patients iron deficiency is usually treated by intravenous route since oral route in HD patients is not efficacious and poorly tolerated.

At the present time there are 2 classes of iron-based phosphate binders: sucroferric oxhydroxide and ferric citrate. Both are effective in the treatment of hyperphosphatemia.

Sucroferric oxhydroxid is poorly absorbed in the gut functioning as a real binder, while ferric citrate is absorbed in the gut also improving iron parameters. Ferric citrate shows a dual functionality controlling serum phosphate and increasing ferritin levels and transferrin saturation. This effect on iron parameters is not observed with sucroferric oxhydroxide.

This study is a 12-month phase IV, multicenter, open-label study that has enrolled 202 patients. It has verified that ferric citrate is effective as phosphate binder but also improved iron parameters (ferritin and TSAT). This effect was only found in patients with low Ferritin levels (≤500 ng/mL) an TSAT (<30%) at baseline.

We have at present 2 iron-based phosphate binders with different properties regarding iron supplementation that may be taken in consideration based on the iron state of the patient.

Ferric citrate is an interesting option for treating both hyperphosphatemia and anemia. It improved hemoglobin levels, TSAT and ferritin with a reduced need for erythropoiesis stimulating agents and intravenous iron. Diarrhea was the most common adverse reaction.

Phosphate binding is a chronic treatment. Is there a need for surveillance of iron overload in patients treated with Ferric Citrate?

What are the mechanisms that avoid iron absorption with sucoferric oxhydroxide?

What are the mechanisms that explain the amelioration of iron parameters only in the patients with low ferritin levels and low transferrin saturation?

Reviewer #3: The primary objective of this 12-month, Phase IV, multicenter, open-label study is to explore the long-term safety and efficacy of ferric citrate in hemodialysis patients in Taiwan, and further evaluate the iron repletion effect and change of iron parameters by different baseline groups. Although this is an interesting study, there are several major trial design and statistical concerns.

Critiques

1. The Abstract should only report the primary and secondary pre-determined endpoints in the protocol or https://clinicaltrials.gov/ct2/show/NCT03256838, i.e., number of subjects with treatment-emergent adverse events (TEAEs), percentage of subjects with treatment-emergent adverse events (TEAEs), and serum phosphorus. Please clearly state which of these are the primary and secondary endpoints. The rest of the analyses should be exploratory, and the results from these analyses should not be included in the Abstract.

2. Please provide either the power analysis or precision analysis in the revised manuscript. In the power analysis, please clearly specify the hypothesis, type I error (one or two-sided), study power, clinical significance level with justification, and the statistical method for determining the study sample size.

3. The authors used the paired t-test to examine the trend effect. Please use the mixed effects model to examine the trend effect because each subject has multiple measurements. The mixed effects model should also adjust for confounding variables. Please provide a detailed multivariable data analysis plan in the revised manuscript that includes (1) model assumptions checking, (2) model performance evaluation, (3) the strategy of handling non-linear terms, (4) the methods of analyzing the missing data, and (5) the method of analyzing interaction terms.

4. All the conclusions, e.g., “the iron-repletion by ferric citrate is effective and gradual, and the increase is limited in patients with a higher baseline,” should be supported by statistical tests with 95% confidence intervals.

5. The authors should conduct a sensitivity analysis for 117 subjects (57.9%) who had completed all the scheduled ferric citrate treatment for 12 months.

6. The authors should discuss the similarities or differences in the results and baseline distributions between this Phase IV trial and previously published Phase III trials.

7. All the tables should include the sample size.

8. Please add the original trial protocol to the Supplemental Information section.

6. PLOS authors have the option to publish the peer review history of their article (what does this mean?). If published, this will include your full peer review and any attached files.

Reviewer #1: No

Reviewer #2: **Yes: **Teresa Adragao

Reviewer #3: No

---

## [Author Response · Author response to Decision Letter 0]

10 Dec 2021

Response letter to PONE-D-21-25691

Reviewer #1: Lee et al. examined the long-term safety and efficacy of ferric citrate in patients undergoing hemodialysis. They showed that ferric citrate efficaciously and safely decreased serum phosphate levels in a prospective, multi-center, open-label, one-year study. 

1. There are a lot of previous studies showing that ferric citrate efficaciously and safely decreased serum phosphate levels. However, many of them have not been cited in this manuscript. The novelty of the present study is questionable.

Response: The clinical effects of ferric citrate in patients with renal disease are received significant attention, with evidence from a number of clinical trials and retrospective studies. In addition to the previous studies mentioned in our article studying ferric citrate for the treatment of hyerpohsphoatemia, Tadashi Yoshida investigated the efficacy and safety of ferric citrate in 33 hemodialysis patients (Int Urol Nephrol. 2021; Jul 15). Some recent research are studying ferric citrate for the treatment of iron deficiency anemia. Pergola et al. recently reported the efficacy and safety of a 12-month ferric citrate treatment in non-dialysis-dependent chronic kidney disease (CKD) (Am J Nephrol. 2021;52(7):572-581). The phase 3 non-inferiority study by Komatsu et al. explored the safety and efficacy of ferric citrate in Japanese patients with iron deficiency anemia (Int J Hematol. 2021 Jul;114(1):8-17). Although these studies continue to confirm that ferric citrate is an effective phosphate binder in different patient groups, the pharmacologic effect of iron repletion on different basal ferritin and TSAT levels are inadequate.

The previous studies with long term results were from Japanese and US population. However, the iron status of Japanese and US dialysis patients represents two extreme iron status groups. In Japan, the average serum ferritin level was 73 ng/ml while it was 887 ng/ml in the US. There are no studies to evaluate its long-term effect in other ESRD patient groups of most of other countries (such as European, Chinese or Korean populations) having ferritin (300~500 ng/ml) at medium level. Therefore, the remaining questions to be explored is what is the suitable practice for ferric citrate in balancing its phosphate binding and iron supplement effect for the majority of the patients around the world. Our study conducted in Taiwan provides real world data to this unknown question. The results of our study revealed the clinical values for of ferric citrate in ESRD patients with medium (300-500 ng/mL) ferritin levels. 

The study population of our study, Taiwanese ESRD population, provides an interesting study angle to evaluate pharmacology effect of a CKD drug product for other Asian countries. Firstly, Taiwan has highest ESRD prevalence rate in the world for many years, and secondly, the dialysis program is fully reimbursed by its healthcare budget. That means 100% coverage with sophisticated management and monitoring system to have a comprehensive view on drug products’ true response. In Taiwan, the iron contents are maintained at a ferritin level of 300−500 ng/mL and TSAT of 30%−50%, which is similar to 2004 European guidance of 200-500 ng/ml and 30%-40% for ferritin and TSAT, respectively. As the diet habits are similar across China, Korea, Taiwan, and several Asian countries, we hope this study could provide a practical direction for ferric citrate’s use in managing both hyperphosphatemia and anemia, the two most prevalent complications in dialysis patients. To facilitate physicians’ decision and prediction on the pharmacological reactions, we further conduct several sub-group analyses by different baseline characteristics after administration of ferric citrate. For example, the changes of ferritin and TSAT in patients with different baseline levels that is commonly used in evaluating iron supplementation. The changes of serum phosphorus in patients with different level to provide more optimized starting dose and lower pill burden.

As this is a Phase IV study to evaluate the effect in the real-world settings, the non-limiting use on the ESAs and IV iron actually provide more insight on the impact of anemia management. We believe that our study may bring useful information to answer following questions (1) While managing serum phosphorus in the target range, could the use of ferric citrate also benefit iron stores in these majority of ESRD patients having mid rang ferritin level? (2) What’s the impact on the existed anemia treatments such as ESAs and IV iron uses? (3) What is the long-term safety of this drug product?

Note: 

According to UK renal registry 20th annual report, the median ferritin in HD patients was 410 mg/dL. [1] 

In a prospective observational study of Korean Clinical Research Center (CRC) for End-Stage Renal Disease (ESRD), which included dialysis patients enrolled from September 1, 2008, at 31 centers in Korea.. The mean ferritin of these Korean patients is 275.5 ng/mL [2]

[1] Rhodri Pyart et al. UK Renal Registry 20th Annual Report: Chapter 7 Haemoglobin, Ferritin and Erythropoietin in UK Adult Dialysis Patients in 2016: National and Centre-specific Analyses. Nephron. 2018;139(suppl1):165–190. doi: 10.1159/000490965

[2] Kwon O et al. Clinical Research Center for End-Stage Renal Disease (CRC- ESRD) Investigators. The Korean Clinical Research Center for End-Stage Renal Disease Study Validates the Association of Hemoglobin and Erythropoiesis-Stimulating Agent Dose with Mortality in Hemodialysis Patients. PLoS One. 2015 Oct 9;10(10): e0140241. doi: 10.1371/journal.pone.0140241. 

2. As described in the Methods section, 5.9% of patients had taken ferric citrate prior to the study. These patients were very low risk of adverse events in the study period. The study has been biased on this point.

Response: We agree with the reviewer’s concern of bias of the 5.9% of patients with a history of using ferric citrate. Thus, this bias concern was added in the limitation section of the revised manuscript (Line 378, Page 16).

3. In the Discussion section, the authors described that “the safety and efficacy analyses are limited to 12 months of treatment; therefore, the conclusions of this study can only be limited within this time frame” as a limitation of this study. Regarding this point, a 3-year retrospective study (Yoshida et al., Int Urol Nephrol, 2021) has been reported recently.

Response: The retrospective study by Yoshida et al. reported the 3-year safety and efficacy of ferric citrate in 33 hemodialysis patients. Although our study has the limitation of 1-year follow-up, our prospective study still shows different clinical values. First and foremost, based on the inclusion criteria of Yoshida’s study, a total of 33 patients received ferric citrate for "at least" 1 month. In addition, patients in that study were co-treated with other phosphate binders that the prescribed dose was smaller than the doses used in other clinical studies. In contrast, in our study, a total of 117 patients completed 12 months of ferric citrate treatment as sole treatment for hyperphosphatemia. Second, the baseline ferritin levels of the Japanese patients in the Yoshida’s study (34 ng/mL) was much more lower and significantly different from patients of most of the world. The Yoshida et al study can only provide conclusion on efficacy and safety in limited patients with a low baseline ferritin level. 

Ferric citrate has dual pharmacological effect on both serum phosphate and iron state. When we use ferric citrate to control serum phosphate level, the impact on the iron state should also be taken into consideration. Serum ferritin is an indicator of body iron stores. As the baseline iron storage is a factor associated with regulation of iron absorption. It is necessary to understand more the effect of ferric citrate in patients with different iron storage level to further explore the impact on the anemia management. 

The baseline ferritin level of our study (353 ng/mL) is much similar to patients of most of the world. Thus, the results of this study can provide more insight on the impact of anemia management in patient with ferritin at medium level. In addition, the post hoc analysis results of this study can help us understand the iron repletion effect of ferric citrate in patients with low (<300 ng/mL), medium (300-500 ng/mL) and high (>500 ng/mL) basal ferritin levels. In addition, it provides a real-world clinical basis for doctors to predicate the ferritin levels during the treatment of ESRD patients with ferric citrate.

4. Please correct the style of references.

Response: Thanks to the referee for pointing out this mistake. In the revised manuscript, the style of references was modified to meet the standard of PLoS One. 

5. Multiple names of adverse events have been written using the capital letters. What do they mean?

Response: We just keep the original terms of the adverse events of the trial protocol, and have no other meanings that we want to emphasize. To avoid unintentional confusion, we amended the capital letters of adverse events in the revised manuscript.

Reviewer #2: 

This study is a 12-month phase IV, multicenter, open-label study that has enrolled 202 patients. It has verified that ferric citrate is effective as phosphate binder but also improved iron parameters (ferritin and TSAT). This effect was only found in patients with low Ferritin levels (≤500 ng/mL) an TSAT (<30%) at baseline.

1. Phosphate binders and iron supplementation are a part of the basic treatment of HD patients. Phosphate binders bind phosphorus in the gut and prevent their absorption. Iron based phosphate binders are a new line of treatment. In HD patients iron deficiency is usually treated by intravenous route since oral route in HD patients is not efficacious and poorly tolerated. At the present time there are 2 classes of iron-based phosphate binders: sucroferric oxhydroxide and ferric citrate. Both are effective in the treatment of hyperphosphatemia. Sucroferric oxhydroxid is poorly absorbed in the gut functioning as a real binder, while ferric citrate is absorbed in the gut also improving iron parameters. Ferric citrate shows a dual functionality controlling serum phosphate and increasing ferritin levels and transferrin saturation. This effect on iron parameters is not observed with sucroferric oxhydroxide. We have at present 2 iron-based phosphate binders with different properties regarding iron supplementation that may be taken in consideration based on the iron state of the patient. Ferric citrate is an interesting option for treating both hyperphosphatemia and anemia. It improved hemoglobin levels, TSAT and ferritin with a reduced need for erythropoiesis stimulating agents and intravenous iron. Diarrhea was the most common adverse reaction. Phosphate binding is a chronic treatment. Is there a need for surveillance of iron overload in patients treated with Ferric Citrate?

Response: This is a very good question. Patients who chronically receive transfusions or use iron-containing medications or supplements (including administered via either iv or oral route) are at risk of iron accumulation and are subject to surveillance of iron overload. Although iron is an essential mineral, excessive iron is harmful for human health. As the iron loading continues, the dedicated iron-binding protein, transferrin, becomes saturated. Excessive iron may damage tissues by catalyzing the formation of reactive oxygen species (ROS) that attach cellular membranes, proteins and DNA, leading to organ damage (Clin J Oncol Nurs. 2009 Oct;13:511-7). Thus, the clinical consequences of iron overload may lead to liver damage, cardiac disease and endocrine disorders. Therefore, it is necessary to assess and monitor the iron status of the patients. On the other hand, since anemia is a common complications of hemodialysis patients and intravenous iron administration is common management of ESRD patients, anemia-related tests (including iron parameters) are routinely performed and monitored in dialysis centers. Therefore, the surveillance of iron overload in patients treated with ferric citrate can be done easily.

2. What are the mechanisms that avoid iron absorption with sucoferric oxhydroxide?

Response: Sucroferric oxyhydroxide is another iron-based phosphate binder recently also approved for the treatment of hyperphosphataemia. Based on the study by Mario Cozzolino et al., sucroferric oxyhydroxide exhibits a low iron-releasing property in the physiological pH range of the gastrointestinal tract. Therefore, the low iron release properties of sucroferric oxyhydroxide support the clinical feature of minimal iron absorption, rather than being caused by avoiding iron absorption.

Reference: Preclinical Pharmacokinetics, Pharmacodynamics and Safety of Sucroferric Oxyhydroxide. Current Drug Metabolism, 2014, 15, 953-965.

3. What are the mechanisms that explain the amelioration of iron parameters only in the patients with low ferritin levels and low transferrin saturation?

Response: The absorption of iron from the intestine into the blood is tightly regulated by iron regulatory proteins and hepcidin/ferroportin system. Hepcidin, a small circulating regulatory hormone peptide, is responsible for regulating the iron homeostasis. Ferroportin is an iron exporter located on the basolateral surface of intestinal enterocytes and macrophages. Hepcidin can bind to ferroportin and induces its internalization and degradation, resulting in cellular iron retention and decreased iron export into circulation. On the other hand, hepcidin levels is negatively regulated by plasma iron concentration and iron stores in the liver (similar to the regulation of glucose by insulin). 

When the body is iron deficient, hepcidin concentrations are low, thereby favoring iron absorption and delivery to the plasma from storage sites; but when the body is iron replete, a higher hepcidin concentration reduces iron absorption and impairs iron release from stores. For patients with relatively low iron storage (ferritin < 500 ng/mL or TSAT< 30% in this study), the body system would be more favor to intestinal iron absorption than those patients with a higher baseline iron storage and it leads to increase serum ferritin levels and TAST level. In contrast, for patients with higher iron storage (ferritin > 500 ng/mL or TSAT>30% in this study), intestinal iron absorption is limited due to systemic iron homeostasis mechanism. We described this in the discussion section of revised manuscript as follows (Line 311-317): Unlike traditional intravenous iron injections, oral ferric citrate is under endogenous regulatory control in the gastrointestinal tract. The regulation from hepcidin and membrane proteins on the small intestine mucosal cell such as DcytB (Ferrireductase), DMT1 (divalent metal transporter1), and Fpn1 (Ferroportin1) tightly control iron’s uptake and mobilization into the blood stream. This mechanism allows iron absorption as needed, and lowers the chance of excessive absorption when the iron stores in the body are sufficient. 

Reviewer #3: The primary objective of this 12-month, Phase IV, multicenter, open-label study is to explore the long-term safety and efficacy of ferric citrate in hemodialysis patients in Taiwan, and further evaluate the iron repletion effect and change of iron parameters by different baseline groups. Although this is an interesting study, there are several major trial design and statistical concerns.

1. The Abstract should only report the primary and secondary pre-determined endpoints in the protocol or https://clinicaltrials.gov/ct2/show/NCT03256838, i.e., number of subjects with treatment-emergent adverse events (TEAEs), percentage of subjects with treatment-emergent adverse events (TEAEs), and serum phosphorus. Please clearly state which of these are the primary and secondary endpoints. The rest of the analyses should be exploratory, and the results from these analyses should not be included in the Abstract.

Response: Thanks for the reviewer’s comments. The primary endpoint was to assess the safety profiles of ferric citrate. The secondary endpoints were to evaluate the efficacy by the time-course changes of serum phosphorus levels and the number of subjects who achieved the target range of serum phosphorus. The primary and secondary endpoints were added in the Abstract of the revised manuscript.

On the other hand, although exploratory outcome measures were not main reason we conducted this study, the results are still very important, especially in the aspect of iron repletion. This is because anemia is the most common complication in ESRD patients. Thus, we removed the description of hemoglobin, aluminum, ALT and AST in the abstract section, but hope to keep the description of result of iron improvement.

2. Please provide either the power analysis or precision analysis in the revised manuscript. In the power analysis, please clearly specify the hypothesis, type I error (one or two-sided), study power, clinical significance level with justification, and the statistical method for determining the study sample size.

Response: Thanks for the reviewer’s suggestion. Since this is a single-arm Phase-IV study, and the major purpose is to collect the real-world effectiveness of ferric citrate as evaluated in an observational, non-interventional trial in a naturalistic setting. In fact, safety profile for a minimum drug exposure in 100 evaluable patients for 6 months and 1 year are required by the Taiwan regulatory authority. Thus, recruiting 200 subjects should be sufficient to collect a 1-year safety profile of at least 100 patients, and the final sample size of 200 subjects is driven by a post-approval commitment to the regulatory authority. 

However, our post-power analysis also supports that the sample size of 200 participants is large enough for this study. We performed the post hoc power analysis to evaluate whether the sample size was sufficient to detect time trend of study outcomes across 13 measurements among hemodialysis patients receiving ferric citrate treatment. The power analysis was based on repeated ANOVA with one group (i.e., all patients received ferric citrate medication) and 13 measurements. The significance level was 0.05 and total sample size was 197. The variance explained by the time effect was 2.48 and the error variance was 1.00. The Greenhouse-Geisser Epsilon was 0.84 for nonsphericity correction. The estimated power was higher than 0.99, indicating that the sample size as well as parameters assumed and given statistics were sufficient to detect the time effect of serum phosphorus across 13 measurements.

3. The authors used the paired t-test to examine the trend effect. Please use the mixed effects model to examine the trend effect because each subject has multiple measurements. The mixed effects model should also adjust for confounding variables. Please provide a detailed multivariable data analysis plan in the revised manuscript that includes (1) model assumptions checking, (2) model performance evaluation, (3) the strategy of handling non-linear terms, (4) the methods of analyzing the missing data, and (5) the method of analyzing interaction terms.

Response: Thanks for the reviewer’s comment. According to reviewer’s suggestion, mixed effects model for primary outcomes were performed to deal with within-subject correlation given patients being followed repeatedly. Statistical assumption, model specifications, and main results were presented.

First, we checked if laboratory tests followed normal distribution by using Kolmogorov-Smirnov test, a method to examine normal assumption. If normal assumption was violated, linear models would not be appropriate for estimation of efficacy of ferric citrate. The results showed that all laboratory measures violated normal assumption, except for hemoglobin. Therefore, all measures were log transformed for subsequent regression analyses, which was a common method of transforming skewed data to conform to normality. After log transformation, the distribution of all laboratory tests, but not TSAT and ferritin, were normally distributed. As a result, all linear regression models were performed by using transformed data.

Second, to evaluate the time trend of study outcomes, non-linear relationship was assessed by adding quadratic term in the mixed effects model. If the quadratic term is statistically significant, there exists a non-linear trend (e.g., a U-shape or an inverted U-shape). Table 1 shows that there were non-linear trends for hemoglobin, serum iron, ferritin, TSAT, and iPTH (all inverted U-shape across time). However, given that the magnitude of coefficients of quadratic form for other lab measures is not obvious despite statistical significance, linear trend would be reported for all study outcomes. 

Table 1. Testing for non-linear trend of study outcomes

[Based on the regulations of PLOS ONE, only text can be filled in this part. Please download the detailed response letter that we uploaded to PLOS ONE]

Third, multivariate mixed effects analyses were also performed with adjustment for study sites, age, sex, and body weight at baseline. Slope was used to represent the time trend of study outcomes; a positive value indicates an increasing trend while a negative value indicates a decreasing trend. The estimate slopes of time trend for study outcomes performed by multivariate mixed effects regression model were presented in Table 2, adjustment for study sites, age, sex, and body weight. Serum phosphorus (slope=-0.004; 95% CI=-0.0076, 0.0002; p=0.042), TIBC (slope=-0.008; 95% CI=-0.0098, -0.0066; p<0.001), and serum calcium (slope=-0.002; 95% CI=-0.0029, -0.0009; p<0.001) had decreasing trend after baseline. There was no linear trend for hemoglobin (slope=0.001; 95% CI=-0.0011,0.0021; p=0.542). Other measures increased across time (i.e., serum iron, ferritin, TSAT, and iPTH) (Table 2).

Table 2. Multivariate mixed effects regression analysis for study outcomes

[Based on the regulations of PLOS ONE, only text can be filled in this part. Please download the detailed response letter that we uploaded to PLOS ONE]

Forth, to assess the influence of missing data due to incomplete follow-ups, we performed a sensitivity analysis by including patients with complete follow-ups, that is, patients who completed 13 visits in the trial. The results were presented in Table 3. The estimated mean and standard error for each laboratory measure at each visit did not change by excluding patients without complete follow-ups (Table 3). Sensitivity analysis for linear trend estimation was also done by including patients with complete follow-ups. The magnitude of estimates for slopes and standard errors was similar to those in our primary analyses. Time trend for serum phosphorus became statistical insignificance (slope= -0.003; 95% CI=-0.0071,0.0011; p=0.157) (Table 4). These results suggested that our main results were not influenced by the missingness due to incomplete follow-ups.

On the other hand, we did not assess group-by-time interactions due to one-arm study design in the current study.

Table 3. Sensitivity analysis for changes in serum phosphorus, hemoglobin, iron-related parameters, calcium and iPTH in patients with complete follow-up (n=117)

[Based on the regulations of PLOS ONE, only text can be filled in this part. Please download the detailed response letter that we uploaded to PLOS ONE]

Table 4. Multivariate mixed effects regression analysis for study outcomes in patients with complete follow-up (n=117)

[Based on the regulations of PLOS ONE, only text can be filled in this part. Please download the detailed response letter that we uploaded to PLOS ONE]

4. All the conclusions, e.g., “the iron-repletion by ferric citrate is effective and gradual, and the increase is limited in patients with a higher baseline,” should be supported by statistical tests with 95% confidence intervals.

Response: Regarding the reviewer’s concern, linear trend estimation was performed to assess the time trend of the 3 ferritin levels. The results also support our previous statistical analysis, showing that ferritin < 300 ng/ mL (95% CI=0.0342, 0.0970) and ferritin 300-500 ng/mL (95% CI=0.0128, 0.0354) had the same significant increase trend after baseline (P<0.001). In addition, the linear trend was not significant over time for ferritin > 500 ng/mL (95% CI=-0.0086, 0.00228; P=0.374).

[Based on the regulations of PLOS ONE, only text can be filled in this part. Please download the detailed response letter that we uploaded to PLOS ONE]

5. The authors should conduct a sensitivity analysis for 117 subjects (57.9%) who had completed all the scheduled ferric citrate treatment for 12 months.

Response: Thanks for the reviewer’s comment. To assess the influence of missing data due to incomplete follow-ups, sensitivity analysis was performed by including patients with complete follow-ups, that is, patients who completed 13 visits in the trial. After excluding patients without complete follow-ups, the mean and standard error of each laboratory measure at each visit did not change much (below table 3). 

Table 3. Sensitivity analysis for changes in serum phosphorus, hemoglobin, iron-related parameters, calcium and iPTH in patients with complete follow-up (n=117)

[Based on the regulations of PLOS ONE, only text can be filled in this part. Please download the detailed response letter that we uploaded to PLOS ONE]

Sensitivity analysis for linear trend estimation was also done by including patients with complete follow-ups. The magnitude of estimates for slopes and standard errors was similar to those in our primary analyses. Only the time trend for serum phosphorus became statistical insignificance. These results suggested that serum phosphorus did slightly influence by the missingness due to incomplete follow-ups.

Table 4. Multivariate mixed effects regression analysis for study outcomes in patients with complete follow-up (n=117)

[Based on the regulations of PLOS ONE, only text can be filled in this part. Please download the detailed reponse letter that we uploaded to PLOS ONE]

6. The authors should discuss the similarities or differences in the results and baseline distributions between this Phase IV trial and previously published Phase III trials.

Response: Thanks for the reviewer’s comment. In our previous Phase III trial in Taiwanese, a total of 147 ESRD patients received fixed dose of ferric citrate 4 g/day or 6 g/day for 8 weeks (J Nephrol. 2015 Feb;28(1):105-13. doi: 10.1007/s40620-014-0108-6. Epub 2014 May 20). The results demonstrated the effectiveness at a daily dose of 4-6 g/day and both doses were well-tolerated in ESRD patients during the 8-week study period. The most common adverse events were gastrointestinal disorder, mostly discolored feces and diarrhea. After 8 weeks of treatment, serum phosphorus declined significantly, while ferritin levels increased significantly. The current Phase IV study further showed the long-term safety and efficacy of ferric citrate at a dose range of 1.5-6.0 g/day. The serum phosphorus was sustainably controlled in long term, while the increase of iron storage showed a trend of plateau after 3-6 months of treatment. In addition, the current Phase IV study further revealed the different iron repletion effects of ferric citrate in ESRD patients with high, medium and low baseline ferritin levels. Furthermore, our real-world effectiveness of ferric citrate as evaluated in an observational, non-interventional trial in a naturalistic setting showed that long-term use of ferric citrate did greatly reduced the need for erythropoiesis-stimulating agents (ESA) and intravenous iron. 

7. All the tables should include the sample size. 

Response: Thanks for the reviewer’s suggestion. The sample size was added to each tables of the revised manuscript. 

8. Please add the original trial protocol to the Supplemental Information section.

Response: The original protocol of this trial was uploaded as the supplementary information.

---

## [Decision Letter · Decision Letter 1]

16 Feb 2022

Long-term safety and efficacy of ferric citrate in phosphate-lowering and iron-repletion effects among patients with on hemodialysis: a multicenter, open-label, Phase IV trial

PONE-D-21-25691R1

Dear Dr. Der-Cherng Tarng

We’re pleased to inform you that your manuscript has been judged scientifically suitable for publication and will be formally accepted for publication once it meets all outstanding technical requirements.

Kind regards,

Pasqual Barretti, Ph.D., MD

Academic Editor

PLOS ONE

Additional Editor Comments (optional):

After rereading the manuscript and based on the unanimous decision of the reviewers, my decision is "Accept"

Reviewers' comments:

Reviewer's Responses to Questions

**Comments to the Author**

1. If the authors have adequately addressed your comments raised in a previous round of review and you feel that this manuscript is now acceptable for publication, you may indicate that here to bypass the “Comments to the Author” section, enter your conflict of interest statement in the “Confidential to Editor” section, and submit your "Accept" recommendation.

Reviewer #1: All comments have been addressed

Reviewer #2: All comments have been addressed

Reviewer #3: All comments have been addressed

2. Is the manuscript technically sound, and do the data support the conclusions?

Reviewer #1: Yes

Reviewer #2: Yes

Reviewer #3: Yes

3. Has the statistical analysis been performed appropriately and rigorously? 

Reviewer #1: Yes

Reviewer #2: Yes

Reviewer #3: Yes

4. Have the authors made all data underlying the findings in their manuscript fully available?

Reviewer #1: Yes

Reviewer #2: Yes

Reviewer #3: Yes

5. Is the manuscript presented in an intelligible fashion and written in standard English?

Reviewer #1: Yes

Reviewer #2: Yes

Reviewer #3: Yes

6. Review Comments to the Author

Reviewer #1: (No Response)

Reviewer #2: The authors have answered to my questions in a satisfactory way. Ferric citrate is an interesting option for treating both hyperphosphatemia and anemia.

Reviewer #3: The authors have responded well to the statistical issues raised in the previous review. There is no further statistical concern about this revised manuscript.

7. PLOS authors have the option to publish the peer review history of their article (what does this mean?). If published, this will include your full peer review and any attached files.

Reviewer #1: No

Reviewer #2: **Yes: **Teresa Adragao

Reviewer #3: No

---

## [Editor Report · Acceptance letter]

22 Feb 2022

PONE-D-21-25691R1 

Long-term safety and efficacy of ferric citrate in phosphate-lowering and iron-repletion effects among patients with on hemodialysis: a multicenter, open-label, Phase IV trial 

Dear Dr. Tarng:

I'm pleased to inform you that your manuscript has been deemed suitable for publication in PLOS ONE. Congratulations! Your manuscript is now with our production department. 

Kind regards, 

on behalf of

Prof. Pasqual Barretti 

Academic Editor

PLOS ONE